# KiSS-1 Modulation by Epigenetic Agents Improves the Cisplatin Sensitivity of Lung Cancer Cells

**DOI:** 10.3390/ijms25095048

**Published:** 2024-05-06

**Authors:** Giovanni Luca Beretta, Desirè Alampi, Cristina Corno, Nives Carenini, Elisabetta Corna, Paola Perego

**Affiliations:** Molecular Pharmacology Unit, Department of Experimental Oncology, Fondazione IRCCS Istituto Nazionale dei Tumori, 20133 Milan, Italy; dmalampi@gmail.com (D.A.); cristina.corno@istitutotumori.mi.it (C.C.); nives.carenini@istitutotumori.mi.it (N.C.); elisabetta.corna@istitutotumori.mi.it (E.C.)

**Keywords:** Non-Small Cell Lung Cancer, KiSS-1, cisplatin, epigenetic modulators, apoptosis, drug combination

## Abstract

Epigenetic alterations my play a role in the aggressive behavior of Non-Small Cell Lung Cancer (NSCLC). Treatment with the histone deacetylase inhibitor suberoylanilide hydroxamic acid (SAHA, vorinostat) has been reported to interfere with the proliferative and invasive potential of NSCLC cells. In addition, the DNA methyltransferase inhibitor azacytidine (AZA, vidaza) can modulate the levels of the metastasis suppressor KiSS-1. Thus, since cisplatin is still clinically available for NSCLC therapy, the aim of this study was to evaluate drug combinations between cisplatin and SAHA as well as AZA using cisplatin-sensitive H460 and -resistant H460/Pt NSCLC cells in relation to KiSS-1 modulation. An analysis of drug interaction according to the Combination-Index values indicated a more marked synergistic effect when the exposure to SAHA or AZA preceded cisplatin treatment with respect to a simultaneous schedule. A modulation of proteins involved in apoptosis (p53, Bax) was found in both sensitive and resistant cells, and compared to the treatment with epigenetic agents alone, the combination of cisplatin and SAHA or AZA increased apoptosis induction. The epigenetic treatments, both as single agents and in combination, increased the release of KiSS-1. Finally, the exposure of cisplatin-sensitive and -resistant cells to the kisspeptin KP10 enhanced cisplatin induced cell death. The efficacy of the combination of SAHA and cisplatin was tested in vivo after subcutaneous inoculum of parental and resistant cells in immunodeficient mice. A significant tumor volume inhibition was found when mice bearing advanced tumors were treated with the combination of SAHA and cisplatin according to the best schedule identified in cellular studies. These results, together with the available literature, support that epigenetic drugs are amenable for the combination treatment of NSCLC, including patients bearing cisplatin-resistant tumors.

## 1. Introduction

Non-Small Cell Lung Cancer (NSCLC) is characterized by late diagnosis and poor prognosis and accounts for more than 85% of all lung cancers [1]. The aggressiveness of NSCLC associates with driver alterations, including EGF-R mutations or ALK translocations, which have already considered as novel therapeutic targeted options [2,3]. These genetic alterations and the consequent enhancement of multiple pathways implicated in tumor growth and survival correlate with an unfavorable prognosis [4,5,6]. Platinum-based therapy is still employed in advanced disease in patients lacking actionable mutations (i.e., mutant EGF-R and KRAS, as well as ALK fusion), but disease outcome remains unsatisfactory due to drug resistance and metastases [3]. Several factors have been implicated in drug resistance development, including gene mutations and epigenetic changes [7,8]. In spite of the identification of driver alterations exploited as therapeutic targets, no effective strategies have been defined to efficiently target the most aggressive features of NSCLC and the concomitant activation of pathways controlling tumor cell growth and survival [3,9]. Epigenetically regulated mechanisms may modulate tumor clonal plasticity, making it possible to reprogram resistant clones by epigenetic treatment, and in this regard, histone deacetylases (HDAC), as well as DNA methyltransferase (DNMT) inhibitors, appear intriguing [10,11]. In fact, these agents may induce the expression of genes favoring cell death and/or growth inhibition [12,13].

Histone modification is involved in tumor development and progression [14,15,16,17,18]. By removing the acetyl groups from amino-terminal lysine residues of histones, HDAC functions as transcription repressor enzymes stimulating chromatin condensation [16]. In addition to their action on histones, different HDAC isoforms act on non-histone proteins as well, including p53 and tubulin, and in such a way affect DNA transcription and repair as well as metabolism [16]. In humans, HDACs are phylogenetically grouped into four classes comprising 18 related enzymes [16,17]. Among these, the Zn-dependent enzymes HDAC 1–11 are targeted by several inhibitors, including vorinostat (suberoylanilide hydroxamic acid, SAHA) [16]. Changes in HDAC levels impact tumor development and progression by impairing growth and apoptotic pathways, which result in reduced apoptosis induction following drug treatment and, in turn, drug resistance [19,20,21]. Increased expression of specific HDAC is reported in many cancer types [17]. This behavior deregulates cell adhesion and motility favoring tumors metastases and drug resistance [22,23].

The cytidine analogue azacytidine (AZA) is incorporated into the genome of rapidly proliferating cells during the S phase of the cell cycle and acts by inhibiting DNMT [24,25]. A dual dose-dependent mechanism of action subtends AZA antitumor activity, including cytotoxicity at high doses, which is mediated by the formation of covalent DNMT-DNA adducts leading, in turn, to DNA damages, and a DNA-hypomethylating effect at lower doses [24]. In addition to the antitumor effects dependent on DNMT inhibition, AZA proved to revert the pro-metastatic phenotype in NSCLC cell lines, suggesting a role for this DNMT inhibitor in preventing metastasis formation [26]. Additionally, therapeutic benefits are provided by the combination of low dose of AZA with cisplatin in NSCLC cell lines, which resulted effective in enhancing the cytotoxic antitumor activity of the platinum-based agent [27].

Epigenetic modifications, such as DNA methylation, histone acetylation, as well as regulation by microRNAs, have been involved in the physiological regulation of KiSS-1 gene expression [28]. The KiSS-1 gene encodes a 145-amino acid polypeptide, which is unstable and is rapidly processed by furin into smaller biologically active peptides called kisspeptins (KPs) [29,30]. Among these, KP-54, also called metastin, is the first peptide generated that is further cleaved to form KP-10, -13, and -14. The mechanism of metastasis suppression involving KiSS-1 has been widely characterized in melanoma [31], in which it has been shown that KiSS-1 needs to be secreted to display a biological function, but the interaction with its receptor, the G-protein coupled receptor-54 (GPR54), does not seem to be necessary. Conversely, in other model systems, GPR54 appears to play critical functions for the anti-metastatic effect [32,33,34]. The activation of KiSS-1 signalling has been associated to inhibition of cancer cells invasion, metastasis formation and tumor recurrence [35]. In addition to its role as metastasis suppressor, KiSS-1/GPR54 has been involved in regulation of drug sensitivity. For instance, the reconstitution of KiSS-1 in cisplatin-resistant head and neck cancer cells restores platinum sensitivity [36], and this feature appears to associate to the pathways activated following KPs-GPR54 interaction, including the activation of kinases. The MAPKs acting downstream of GPR54 in tumor cells play both protective and pro-apoptotic roles depending on the molecular context [37]. More recently, KiSS-1 has been reported to play a role in modulating the apoptotic response of melanoma cells to antitumor drug exposure, including vemurafenib [38]. This feature allows for the speculation that the combination of antitumor therapeutics with KPs may improve patients’ response to multiple therapeutic agents [38].

In NSCLC, a role for KiSS-1 in metastasis suppression and cisplatin sensitivity has been hypothesized [21,30,39]. A recent study supports an inverse relationship between KiSS-1 expression and tumor progression, since stage I–II NSCLCs showed higher levels of KiSS-1 than III-IV NSCLCs [30]. Differences of KiSS-1 expression levels between stage III and IV have also been detected [30]. Moreover, KiSS-1 expression was higher in NSCLC primary tumors compared to metastatic sites, suggesting a role for KiSS-1 as metastasis suppressor in NSCLC [30]. In addition, the observation that the expression of KiSS-1 is down-regulated in NSCLC cells selected for resistance to cisplatin and displaying enhanced metastatic behavior suggests an association between platinum sensitivity and KiSS-1 levels [21,39].

Here, we sought to explore drug combinations involving cisplatin and epigenetic drugs SAHA or AZA and to investigate cell response to treatment, taking advantage of the availability of NSCLC cells characterized by acquired platinum drug resistance. Moreover, since KiSS-1 is up-regulated by the treatment of NSCLC with HDAC inhibitors [21], we analyzed KiSS-1-derived peptides in supernatants of cells treated with the drug combinations, as well as the effect of exogenous KPs on sensitivity to cisplatin. Finally, in vivo studies performed in mice bearing H460 and H460/Pt tumor xenografts treated with SAHA, cisplatin, and the combination are discussed.

## 2. Results

### 2.1. Sensitivity of NSCLC Cell Lines to Cisplatin, SAHA and AZA

Cell sensitivity to cisplatin, SAHA and AZA was evaluated using growth inhibition assay on H460 and H460/Pt cells. Twenty-four hours after seeding, cells were exposed to cisplatin for 1 and 72 h or to SAHA and AZA for 72 h (Table 1). Compared to H460, H460/Pt cells were 3- and 2.4-fold resistant to cisplatin at 1 and 72 h, respectively. The two cell lines showed similar sensitivity to SAHA, whereas a slight reduction in sensitivity to AZA was observed in resistant with respect to sensitive cells.

### 2.2. Drug Combination Studies

Two different treatment schedules were considered for the drug combination of cisplatin with SAHA or AZA. Specifically, cells were simultaneously exposed to the drugs for 72 h or preincubated with SAHA or AZA for 24 h before the co-treatment with cisplatin for 48 h. Regarding the concomitant exposure to cisplatin and SAHA on H460 cells, the drug interaction observed was dependent on the concentrations. The CI values displayed an antagonistic effect at the concentrations of 10 µM cisplatin combined with 0.3, 1, or 3 µM SAHA (Table 2, Appendix A).

Under other concentrations, the CI values ranged between 0.56 and 1.56, with some experimental points lower than 0.7, which is indicative of synergism. When H460 cells were simultaneously treated with the combination of AZA and cisplatin, a favorable drug interaction (CI values around 0.4) was observed with 3 and 1 µM cisplatin concentrations in combination with the higher concentration of AZA, or with the high concentrations of cisplatin (10, 3 and 1 µM) in combination with the lower concentration of AZA. When cells were preincubated with SAHA, no antagonistic interaction was evident in the combination with 10 µM cisplatin and SAHA (CI 0.82–1.06). CI values lower than 0.7 were observed under higher SAHA concentrations (Table 3, Appendix A). Compared to simultaneous treatment, a more favorable pattern of drug interaction was found when cells were pre-treated with AZA (CI values lower than 1 in all the conditions tested). Specifically, a strong synergism was observed in the combination of 10, 3, and 1 µM cisplatin with 1 and 0.3 µM AZA.

In cisplatin-resistant cells, the concomitant exposure to SAHA and cisplatin did not result in a favorable drug interaction (CI values 1.07–2.34, Table 4, Appendix A). This finding was also observed following the simultaneous exposure of AZA and cisplatin. As reported in Table 4, only a few experimental points resulted in marked synergism (1 µM AZA plus 3 µM cisplatin; 0.3 µM AZA plus 10 or 3 µM cisplatin).

When H460/Pt cells were pre-incubated with SAHA, the overall drug interaction appeared to be more favorable than in concomitant exposure with some experimental point displaying CI values lower than 1 (Table 5, Appendix A). This finding was observed also for the preincubation with AZA. Indeed, when using 1 µM, AZA a synergistic interaction was evident at most used cisplatin concentrations (CI values all lower than 1). At 0.3 µM AZA, three of the five tested cisplatin concentrations resulted in a marked synergistic interaction.

### 2.3. Modulation of Proteins Involved in Apoptosis Induction

The modulation of the proteins implicated in apoptotic cell death was examined in H460 and H460/Pt cells treated with the combination of SAHA or AZA with cisplatin. To this end, proteins that are known to play a major role in drug-induced apoptosis were selected. Cells were exposed to different concentrations of SAHA (0.3, 1, and 3 µM) or AZA (0.1, 0.3, and 1 µM) for 24 h, and then cisplatin was added for 48 h. The concentration of cisplatin used for H460 cells was 3 µM, whereas H460/Pt cells were exposed to 10 µM. Cisplatin exposure up-regulated p53 levels both in sensitive and in resistant cells when used at 3 and 10 µM, respectively. This behavior was maintained following the combination with SAHA, which per se already slightly increased p53 levels (Figure 1).

The pattern of modulation of Bax levels was similar to that observed for p53 in both cell systems. Conversely, no important modulation of Bcl2 levels was evidenced in parental and resistant cells. Regarding the combination of cisplatin with AZA, in H460 cells, the most striking effect on p53 levels was observed when cells were treated with the combination of 1 or 0.3 µM AZA and cisplatin. Moreover, Bax levels were increased after cisplatin exposure alone and upon combined treatment. A marked up-regulation of p53 levels was evidenced in H460/Pt cells upon cisplatin treatment, and this modulation was maintained following the combination with different AZA concentrations. The pattern of up-regulation observed for the pro-apoptotic protein Bax was similar to that observed for p53, with already a strong increase as compared to control cells for those treated with cisplatin alone. Under this experimental condition, the levels of Bcl2 were not importantly altered by treatment both in parental and resistant cells. Such results support that an enhanced induction of p53 can be achieved when combining low concentrations of SAHA or AZA with cisplatin, therby suggesting an involvement of apoptosis in the synergistic effects.

The induction of apoptosis in H460 and H460/Pt cells upon treatment with the combination of SAHA or AZA and cisplatin was carried out (Figure 2 and Figure 3). An increased number of apoptotic cells was observed in H460 cells exposed to the combination of 3 μM SAHA and 3 μM cisplatin, as compared to cisplatin or SAHA alone. A similar behavior was found in the resistant variant when using 10 μM cisplatin and 3 μM SAHA (Figure 2). These findings indicate that SAHA can increase cell susceptibility to cisplatin-induced apoptosis.

An increased number of apoptotic cells was also observed in H460 cells exposed to the combination of 0.3 or 1 μM AZA and 3 μM cisplatin, as compared to cisplatin or AZA alone. This behavior was found in the resistant variant when using 10 μM cisplatin and 0.3 or 1 μM AZA (Figure 3). Such a behavior is consistent with the biochemical analysis of the apoptotic response reported above.

### 2.4. KiSS-1 Modulation upon Drug Treatment

ELISA was used to quantitatively assess KiSS-1 levels in cell media from H460 and H460/Pt cells. Specifically, cells were exposed to SAHA (0.3, 1, and 3 µM) or AZA (0.1, 0.3, and 1 µM) for 24 h and then co-incubated with cisplatin (3 µM for H460 and 10 µM for H460/Pt cells) for 48 h, and the levels of KiSS-1 released into the culture medium were measured. When SAHA was used in H460 and H460/Pt cells in combination with cisplatin (0.3 µM SAHA plus 3 µM cisplatin or 0.3 µM SAHA plus 10 µM cisplatin), no significant changes in KiSS-1 release were observed in comparison to the control untreated cells (Figure 4). Conversely, with respect to the control cells, higher levels of KiSS-1 were released in the culture medium of both parental and cisplatin-resistant cells upon exposure to 1 µM AZA and cisplatin (3 µM for H460 and 10 µM for H460/Pt cells). Regarding H460/Pt cells, significant increased KiSS-1 levels were found in the comparison between control untreated cells or cells exposed to 0.1 µM AZA and 1 or 0.1 µM AZA plus 10 µM cisplatin. Moreover, significantly increased KiSS-1 levels were observed in the comparison between 10 µM cisplatin and its combination with 1 or 0.1 µM AZA. Taken together, these results support the possibility to influence KiSS-1 releae with pharmacological treatment. The released KiSS-1 may act as a modulator of cisplatin-induced apoptosis.

### 2.5. Effect of Exogenous KiSS-1 on Cisplatin-Induced Apoptosis

To futhere investigate the contribution of KiSS-1 to the modulation of cisplatin sensitivity, we examined the effect of an exogenous KiSS-1-derived peptide, specifically KP10, on cisplatin-induced apoptosis in H460 and H460/Pt cells (Figure 5). Indeed, the KiSS-1 polypeptide derived from transcription of the KiSS-1 gene is cleaved into biologically active peptides known as kisspeptines. No apoptotis induction was evidenced upon exposure of H460 and H460/Pt cells to 500 and 5 ng/mL KP10. The concomitant treatment with cisplatin and KP10 markedly increased the apoptosis induction in both cell lines. This effect was particularly evident when cells were exposed to the combination cisplatin and 500 ng/mL KP10. These results support a sensitizing role for KiSS-1 to the cytotoxic effect of cisplatin.

### 2.6. Antitumor Activity

The effects of SAHA given orally on the antitumor efficacy of i.v. cisplatin were tested in nude mice bearing s.c. human lung carcinomas. The treatment started 5–8 days after cell injection, when tumors were established. A suboptimal dose level of cisplatin (4.5 mg/kg) was employed in order to more easily assess a synergism with SAHA. Thus, negligible antitumor effects were achieved by cisplatin delivered alone to mice bearing both H460 and H460/Pt tumors (Figure 6). Likewise, no significant results were achieved by SAHA on the H460 model. Conversely, the H460/Pt variant moderately responded to the HDAC inhibitor, with a TVI of 35% (*p* < 0.05). Upon SAHA pretreatment, a marked inhibition of tumor growth was found both in the parental and in the resistant model in terms of TVI (57 and 65%). Compared to the treatment with SAHA or cisplatin alone, a more lasting effect on the growth of the tumors was achieved by the combination. Thus, SAHA was able to significantly enhance the activity of cisplatin against lung malignancies derived from two model systems characterized in vitro by responsiveness or acquired resistance to the drug and in vivo by a quick or slow growth in the host.

## 3. Discussion

In spite of the advancements in the molecular characterisation and identification of vulnerabilities in NSCLC, there is still an urgent need to define novel treatment options for this disease, since persistent cures cannot be achieved.

This study was conceived to examine if combined treatment with epigenetic agents, i.e., HDAC or DNMT inhibitors, could improve the anti-proliferative effect of cisplatin and whether the antitumor efficacy of the combination could be related to the metastasis suppressor KiSS-1, that has been recently implicated by us and others in drug-sensitizing effects [38,39,40]. For this purpose, both cisplatin-sensitive H460 and -resistant H460/Pt cells were employed. Given the relevance of apoptosis in cell response to cisplatin, we analyzed the modulation of proteins implicated in this type of cell death. In addition, since it has been previously shown that the treatment of the cell lines employed here with HDAC inhibitors resulted in up-regulation of the mRNA levels of the metastasis suppressor KiSS-1 [21], we also examined the KiSS-1 levels in our experimental models after drug exposure. H460/Pt cells are characterized by a stable resistance to cisplatin linked to a reduced susceptibility to apoptosis. Both parental and resistant cell lines exhibit wild-type p53 and *KRAS* mutation. The employed epigenetic drugs were effective in inhibiting cell proliferation in the two cell lines, with IC_50_ values in the low micromolar range. Particularly, a marked and similar anti-proliferative effect of SAHA was observed both in sensitive and resistant cells. The IC_50_ values of AZA were lower than those of SAHA in H460 and H460/Pt cells, with the mean IC_50_ value of resistant cells higher than that of H460 cells. An analysis of the effect of the combination of cisplatin and SAHA or AZA using different schedules of treatment and a range of concentrations endowed with different levels of cytotoxicity, to allow the use of robust methods for drug interaction assessment (i.e., Chou and Talalay), indicated that a favorable combination was achieved when cells were treated with the epigenetic drug 24 h before the addition of cisplatin. Such an effect was clearly synergistic for the cisplatin-AZA combination. When AZA was used as modulating agent, the efficacy of the combination was higher upon pre-incubation with AZA as compared to the simultaneous treatment. For example, a CI value of 0.27, supporting a synergistic interaction, was found in H460 cells in the former condition at 10 µM cisplatin with 1 µM AZA, whereas in the latter, the value was 0.91, indicating additivity. Similarly, in the resistant variant, a CI value of 0.58 was observed upon simultaneous incubation with 10 µM cisplatin and 1 µM AZA, whereas the value was 0.16 when AZA was added before cisplatin. These findings suggest that the treatment with epigenetic agents may prime cells to respond to the cytotoxic cisplatin by favoring its anti-proliferative activity. An improved antitumor activity was indeed observed when testing the combination of cisplatin and SAHA in mice xenografted with parental and resistant cells.

To clarify the mechanisms involved in the observed favorable drug interactions, we examined the modulation of the levels of selected proteins implicated in apoptosis, focusing on those that are known to play a major role in drug-induced apoptosis. Western blotting indicated that SAHA *per se* up-regulated p53 levels in both H460 and H460/Pt cells when used at 3 µM. As expected, cisplatin at cytotoxic concentrations (3 µM in parental cells and 10 µM in resistant cells) also increased p53 levels. This behavior was corroborated by the induction of apoptotic cell death. Combined treatment with cisplatin and 1 or 0.3 µM SAHA again resulted in enhanced p53 levels. In keeping with p53 up-modulation, we detected an increase of the levels of the pro-apoptotic protein Bax, encoded by a p53 target gene. When considering the effects of AZA alone or in combination with cisplatin on the modulation of apoptosis-related proteins in parental cells, we observed an up-regulation of p53 already with single agents, specifically with 1 and 0.3 µM AZA and with 3 µM cisplatin. A much stronger induction of p53 levels was found when these concentrations of AZA were combined with cisplatin. AZA per se displayed a modest effect on p53 levels in resistant cells, whereas a marked enhancement of p53 was observed upon treatment with cisplatin alone or with all the drug combinations. This finding is in line with the observation that the exogenous introduction of p53 in NCI-H1299 cells enhanced their sensitivity to taxol via the activation of intrinsic apoptotic death pathway by stimulating caspase-3 and -9 activation [41]. An up-regulation of the pro-apoptotic protein Bax was observed in both cell systems following drug combination, suggesting apoptotic cell death. The extent of modulation observed for the anti-apoptotic protein Bcl2 was marginal.

A relevant observation of this study was the capability of epigenetic agents to enhance cisplatin- induced apoptosis.

Since KiSS-1 has been reported to favor apoptosis in specific cellular contexts [38,40], we carried out an analysis of KiSS-1 release in the culture medium upon treatment with the different drugs and drug combinations. Because *KiSS-1* encodes an unstable polypeptide that is promptly cleaved into kisspeptins, we used ELISA to quantitatively measure the effect of treatment on KiSS-1-derived peptides. In both parental and resistant cells, the exposure to AZA increased the levels of KiSS-1 peptides in the culture medium of treated as compared to untreated cells. A statistically significant increase was observed for the combination of 1 µM AZA and 3 µM cisplatin in H460 cells and for 1 or 0.1 µM AZA and 10 µM cisplatin in resistant cells. Cell exposure to SAHA indicated that SAHA modulated KiSS-1-derived peptide levels, although we did not find a statistical significance.

We used a loss-of-function approach to examine the impact of KiSS-1 on cispaltin response. No modulation of apoptosis was observed, implying that KiSS-1 per se cannot affect drug response when molecularly inactivated. However, exogenous KPs appeared capable to improve apoptosis. Thus, we extended previous observations from Gatti et al. [39], employing a gain-of-fuction approach based on the use of exogenous kisspeptins, and observed that the exposure to the kisspeptin KP10 improved the cisplatin-induced apoptosis of H460 and H460/Pt cells.

The obtained results suggested a possible chemo-sensitizing role of KiSS-1 to cisplatin. Indeed, KiSS-1 mRNA levels have been shown to be up-regulated by HDAC and DNMT inhibitors [21,42], and KiSS-1 has been proposed to sensitize head and neck squamous cell carcinoma, lung cancer models to cisplatin, and melanoma cells to vemurafenib [38,39,40]. Sensitization appers to be related to enhanced apoptotic response; both AZA and SAHA have been shown to upmodulate KiSS-1 expression [21,39], suggesting that KiSS-1 upmodulation favors apoptosis induction. The mechanim we propose for KiSS-1 role is represented in Figure 7. Future effort will be focused on further exploring the possibility to enhance cisplatin response in animal models by KiSS-1 modulation. Due to the poor bioavailability of KPs, they are not suitable for in vivo studies, but the development of KiSS-1 mimetics endowed with improved stability and bioavailability will be helpful to achieve this objective.

## 4. Materials and Methods

### 4.1. Drugs, Cell Lines, Culture Conditions

Cisplatin (Teva, Italia) was diluted in 0.9% NaCl solution. The suberoylanilide hydroxamic acid, SAHA (Vorinostat, Selleckhem, Aurogene, Rome, Italy), and the pyrimidine nucleoside analogue, 5-Azacytidine (Vidaza, Selleckhem), were diluted in dimethyl sulfoxide (DMSO, Sigma–Aldrich, St. Louis, MO, USA). The final concentration of DMSO in cell culture never exceeded 0.25%. The KiSS1-derived peptide KP10 was from Anaspec, DBA Italia (Milan, Italy).

The NSCLC H460 and the cisplatin-resistant variant H460/Pt cells, which are both characterized by KRAS mutation, were cultured in RPMI1640 medium plus 10% fetal bovine serum. The resistant cells were obtained in our laboratory by exposing H460 cells to increasing concentration of cisplatin. H460/Pt cells displayed a stable drug-resistant phenotype and were characterized by increased invasive ability as well as aggressive behavior. The growth characteristic of the two cell lines were similar and, following the establishment of cisplatin resistance [43], H460/Pt cells were cultured in the absence of cisplatin. Resistance was routinely checked by cell growth inhibition assays.

### 4.2. Growth Inhibition Assay and Drug Interaction Studies

Cell sensitivity was assessed by growth-inhibition assays using a cell counter. Twenty-four hours after seeding, cells were exposed to drugs for 1 or 72 h. In the case of 1 h exposure, the drug-containing medium was replaced with fresh medium, and cells were cultured for additional 71 h. At the end of the treatment, floating cells were removed, and attached cells were trypsinized and counted using coulter counter (Z2 Particle Counter, Beckman Coulter, Milan, Italy). At least three independent experiments were performed. IC_50_ is defined as the concentration of a drug inhibiting 50% of cell growth. The effect of drug combinations was evaluated by exposing the cells to different concentrations of drugs according to the Chou–Talalay method. By applying the Calcusyn software, Ver. 2 (Biosoft, Cambridge, UK), this method assigns a combination index (CI) value to the drug combination. CI values lower than 0.85–0.90 indicate synergistic drug interactions; conversely, CI values higher than 1.20–1.45 or around 1 reflect antagonism or additive activity, respectively.

### 4.3. Western Blot Assays

Standard procedures were applied for the Western blot assay [21]. Cells were harvested by cell dissociation buffer and collected by centrifugation. Harvested cells were lysed in 0.125 M Tris–HCl pH 6.8 (Sigma–Aldrich), 5% sodium dodecyl sulfate (SDS, Lonza, Verviers, Belgium), and protease/phosphatase inhibitors (25 mM sodium fluoride, 10 μg/mL pepstatin A, 1 mM phenylmethylsulfonyl fluoride, 10 μg/mL trypsin inhibitor, 12.5 μg/mL leupeptin, 30 μg/mL aprotinin, 1 mM sodium orthovanadate, and 1 mM sodium molibdate, all purchased from Sigma–Aldrich). Following sonication and boiling, protein content was determined through the bicinchoninic acid (BCA) assay method (Pierce, Thermo Fisher Scientific, Waltham, MA), and SDS-PAGE was performed. Proteins were then transferred onto nitrocellulose membranes. The films were incubated in “Blocking buffer” (PBS-Tween containing 5% (*w*/*v*) dried skimmed milk) for 1 h and incubated overnight at 4 °C with the primary antibodies dissolved in the blocking buffer. Following 1 h incubation at room temperature with horseradish-conjugated secondary antibodies, membranes were exposed to chemiluminescence solution (ECL, GE Healthcare).

### 4.4. Apoptosis Analysis

The Terminal deoxynucleotidyl transferase dUTP Nick End Labelling (TUNEL, Roche, Mannheim, Germany) assay was used to measure late apoptotic events. H460 and H460/Pt cells were exposed to 3, 1, and 0.3 μM SAHA for 24 h and then co-incubated for 48 h with 3 μM (for H460) or 10 μM (for H460/Pt) cisplatin. At the end of treatment, both floating and adherent cells were harvested, washed in PBS, and fixed by 4% paraformaldehyde (Sigma–Aldrich) dissolved in PBS for 45 min at room temperature. After washing with PBS, cells were treated with permeabilising solution containing 0.1% Triton X-100 (Sigma–Aldrich) and 0.1% sodium citrate (Sigma Aldrich) in PBS for 2 min on ice. After washing with PBS, each sample was resuspended in 50 μL of TUNEL reaction mix (e.g., 5 μL of enzyme and 45 μL of label solution). After incubation for 1 h at 37°C in the dark, samples were washed and resuspended in 1 mL PBS and analyzed by flow cytometer (BD Accuri C6, Becton Dickinson, Milan, Italy). Data were analyzed using associated flow cytometer software Ver. 1, Becton Dickinson.

The annexin V-binding assay (Immunostep, Salamanca, Spain) was also used to evaluate apoptosis. H460 and H460/Pt cells were exposed for 48 h to KP10, cisplatin, or their simultaneous combination. At the end of the treatment, cells were washed with cold phosphate-buffered saline and resuspended in binding buffer (10 mM Hepes-NaOH, pH 7.4, 2.5 mM CaCl_2_, and 140 mM NaCl). Cells were then incubated at room temperature in the dark for 15 min with 5 µL of FITC-conjugated annexin V and 10 µL of 2.5 µg/mL propidium iodide. Annexin V-binding was examined by flow cytometry (BD Accuri, Becton Dickinson,), acquiring 10,000 events for sample. Results were analyzed using the associated instrument’s software (Becton Dickinson).

### 4.5. ELISA

KiSS-1 levels released into the culture medium were measured by ELISA (Human Metastasis Suppressor KiSS-1 kit, Cusabio, Houston, TX, USA), according to the manufacturer’s instructions for quantitative analysis. After treatment with cisplatin and SAHA or cisplatin and AZA alone or in combination, the culture supernatants of H460 and H460/Pt cells were collected. To normalize the KiSS-1 peptide levels released, adherent cells were harvested and counted. Three independent experiments were performed, and the mean value ± standard deviation (SD) was calculated.

### 4.6. Antitumor Activity Studies

Female athymic CD-1 nude mice, 8–10 weeks old (Charles River, Calco, Italy), were employed. Mice were maintained in laminar flow rooms, keeping temperature and humidity constant. Mice had free access to food and water. Experiments were authorized by the Italian Ministry of Health according to the national law (Project Identification: 1103/2015-PR.; approved by the Ministry of Health on 10 October 2015) in compliance with international policies and guidelines. The H460 and H460/Pt were used as xenografts in the study. Exponentially growing cells (5 × 10^6^/mouse) were s.c. injected into the right flank on day 0 forming experimental groups of 7–8 animals. Tumor growth was followed by biweekly measurements of tumor diameters with a Vernier caliper. Tumor volume (TV) was calculated according to the formula TV (mm^3^) = d2 × D/2, where d and D are the shortest and the longest diameter, respectively. Treatment started when tumors were established at different times after cell inoculum. SAHA was delivered orally every weekday at a dose of 100 mg/kg followed by cisplatin, 4.5 mg/kg, i.v. 4–5 h after the last SAHA administration every week for 3–4 weeks. SAHA (Selleckhem) was dissolved in DMSO and cremophor ELP and suspended in PBS 1 × (10 + 5 + 85%). Cisplatin (Teva) was ready to use. The compounds were delivered in a volume of 10 mL/kg of body weight.

Tumor volume inhibition percentage (TVI %) in treated (T) versus control (C) mice was calculated as: TVI% = 100 − (mean TV treated/mean TV control × 100).

### 4.7. Statistical Analysis

All experiments were repeated in triplicate. Statistical analysis was performed using GraphPad Prism 5 (GraphPad Software, United States). The unpaired Student’s *t*-test was employed for evaluating significance in difference of means between groups. *p* < 0.05 was considered significant. For in vivo studies, Student’s *t*-test (two-tailed) was used for statistical comparison of tumor volumes in mice.

## 5. Conclusions

Taken together, our results support that epigenetic drugs represent promising agents for the treatment of NSCLC. Both SAHA and AZA are effective in increasing cisplatin sensitivity in cisplatin-sensitive and -resistant cells, and the effect of the combination appears to be dependent on the treatment schedule, as a synergistic interaction occurs when cells are treated with the modulator followed by cisplatin. A biochemical analysis of cell response suggests the activation of apoptotic cell death with modulation of p53 and Bax as well as with increased release of KiSS-1-derived peptides in cell supernatants under specific conditions. Additionally, besides supporting the interest of drug combinations with conventional agents for the treatment of NSCLC, this study suggests that KiSS-1 may contribute to sensitising cells to cisplatin. Additional effort will be required to design KPs mimic endowed with drug-like properties suitable for further in vivo testing in animal models. Overall, by enhancing the sensitivity of NSCLC cells to cisplatin through a mechanism involving KiSS-1, epigenetic drugs appear to be promising for the treatment of NSCLC patients, including those bearing cisplatin-resistant tumors.

## Figures and Tables

**Figure 1 ijms-25-05048-f001:**
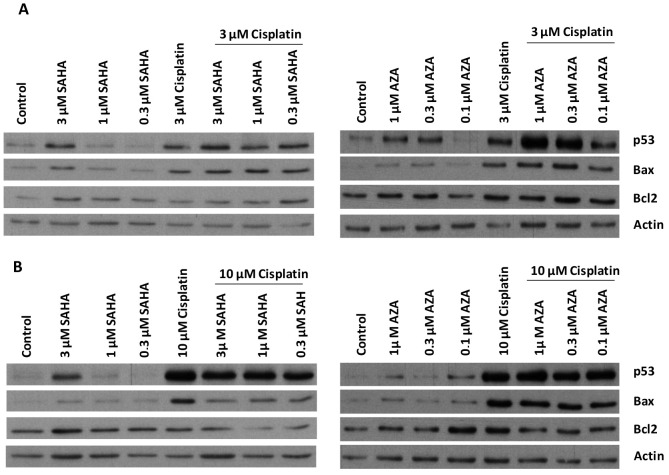
Western blot analysis of the expression of p53, Bax, and Bcl2 in H460 (**A**) and H460/Pt (**B**) cells after the treatment with SAHA or AZA and cisplatin. Cells were exposed to SAHA or AZA for 24 h and then co-incubated with cisplatin for 48 h. Protein expression was analyzed by Western blotting. Actin was used as control for loading. The band intensity was quantified through the program Image Studio Lite Ver.5.2 (LICOR Biosciences, Bad Homburg, Germany). The values obtained were normalized according to the loading control (Actin, Appendix A).

**Figure 2 ijms-25-05048-f002:**
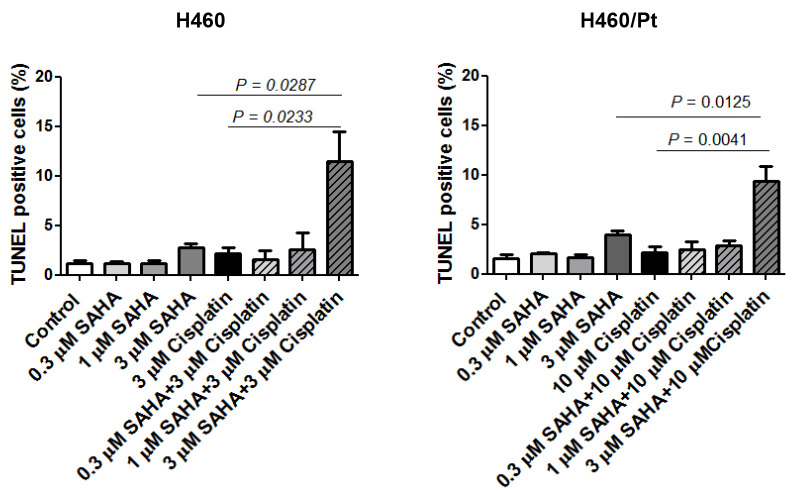
Apoptosis induction in H460 and H460/Pt cells following exposure to SAHA and cisplatin. Cells were treated with SAHA for 24 h and then co-incubated with cisplatin for 48 h. Apoptotic cell death was evaluated at the end of the treatment through a TUNEL assay. The values reported in the graph represent the means of three experiments and the corresponding standard deviations. The statistical analysis was performed by a *t*-test with the software GraphPad Prism, Ver.5 (Boston, MA, USA). *p* < 0.05 were considered significant.

**Figure 3 ijms-25-05048-f003:**
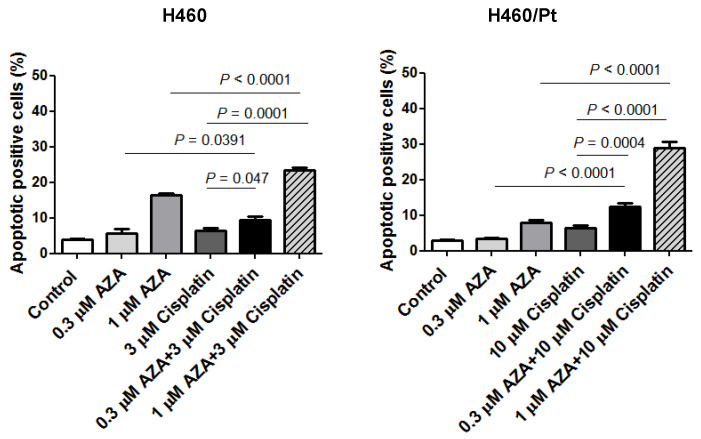
Apoptosis induction in H460 and H460/Pt cells following exposure to AZA and cisplatin. Cells were treated with AZA for 24 h and then co-incubated with cisplatin for 48 h. Apoptotic cell death was evaluated at the end of the treatment through an annexin V-binding assay. The values reported in the graph represent the means of three experiments and the corresponding standard deviations. The statistical analysis was performed by *t*-test with the software GraphPad Prism Ver.5 (Boston, MA, USA). *p* < 0.05 were considered significant.

**Figure 4 ijms-25-05048-f004:**
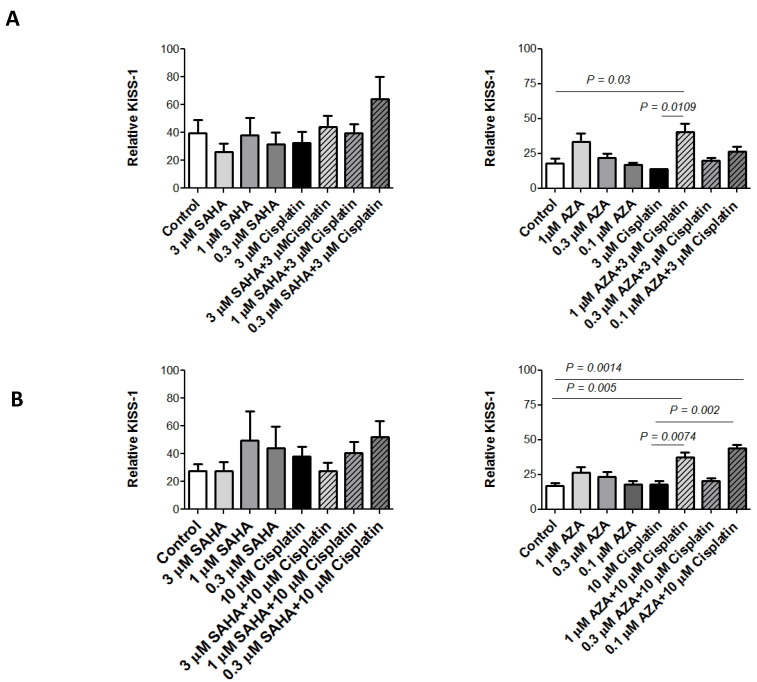
KiSS1 quantification in H460 (**A**) and H460/Pt (**B**) cells after treatment with SAHA or AZA and cisplatin. H460 (**A**) and H460/Pt (**B**) cells were pre-incubated with SAHA or AZA for 24 h and then co-incubated with cisplatin for 48 h. KiSS1 levels were normalized according to cell number. The values reported in the graph represent the means of three experiments and the corresponding standard deviations. The statistical analysis was performed by a *t*-test with the software GraphPad Prism Ver.5 (Boston, MA, USA). *p* < 0.05 were considered significant.

**Figure 5 ijms-25-05048-f005:**
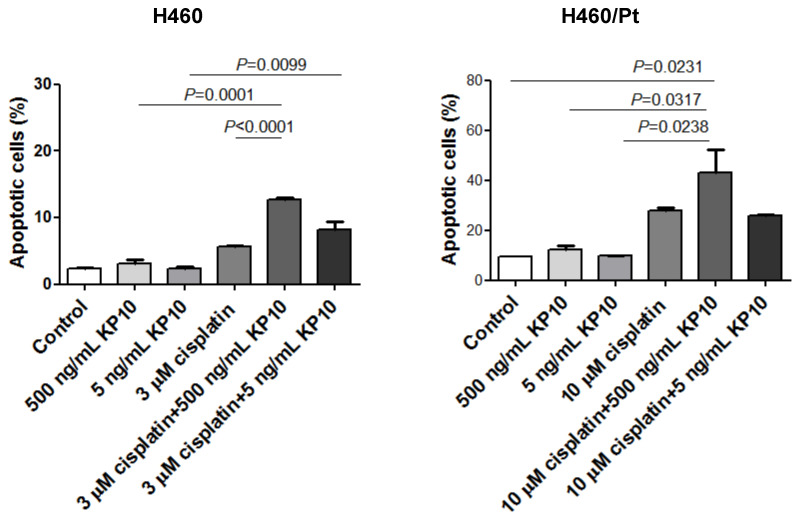
Analysis of apoptosis by annexixn V-binding assay in H460 and H460/Pt cells following exposure to KP10 and cisplatin. Twenty-four hours after seeding, cells were exposed to cisplatin, KP10, or their combination. An annexin V-binding assay was used to measure the apoptosis after 48 h of treatment. The statistical analysis was performed by a *t*-test with the software GraphPad Prism Ver.5 (Boston, MA, USA). *p* < 0.05 were considered significant.

**Figure 6 ijms-25-05048-f006:**
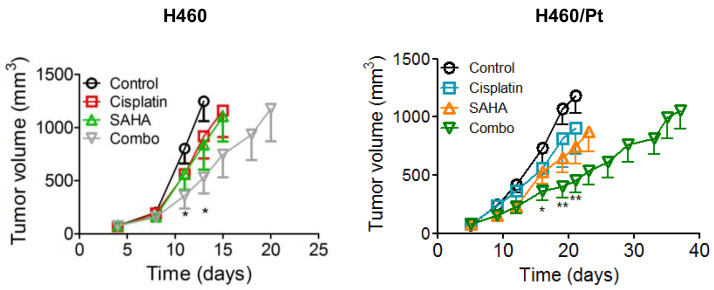
Antitumor activity of SAHA and cisplatin in nude mice bearing human H460 and H460/Pt tumor xenograft. Animals bearing tumors were treated by oral route with 100 mg/kg SAHA every weekday followed by cisplatin at 4.5 mg/kg administered i.v. 4–5 h after the last SAHA administration every week for 3–4 weeks. The statistical analysis was performed by a *t*-test with the software GraphPad Prism Ver.5 (Boston, MA, USA). * *p* < 0.05, ** *p* < 0.005.

**Figure 7 ijms-25-05048-f007:**
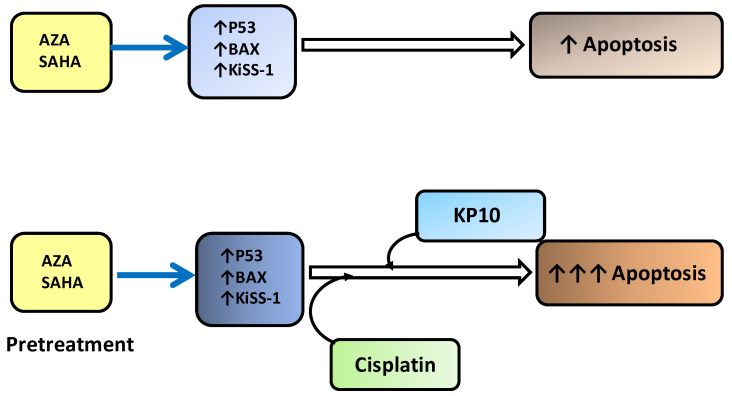
Hypothetical model summarizing the mechanism behind the cellular events supporting the drug combination efficacy.

**Table 1 ijms-25-05048-t001:** Cell sensitivity to cisplatin, SAHA, and AZA ^1^.

Cell Line	Cisplatin (µM, 1 h)	RI	Cisplatin (µM, 72 h)	RI	SAHA (µM, 72 h)	R	AZA (µM, 72 h)	RI
H460	15.46 ± 5.0		1.02 ± 0.4		1.36 ± 0.5		0.50 ± 0.1	
H460/Pt	49.4 ± 11.0	3	2.4 ± 0.9	2.4	1.07 ± 0.1	0.8	0.85 ± 0.1	1.7

^1^ Cell sensitivity to drugs was evaluated by growth inhibition assays based on cell counting. Exponentially growing cells were seeded, and 24 h later, they were exposed to drugs for 1 h or 72 h. Cells were counted with a cell counter 72 h after the treatment started. IC_50_ represents the drug concentration inhibiting cell growth by 50%. RI represents the Resistant Index, which is defined as the ratio between the IC_50_ of resistant and sensitive cells. The values reported in the table represent the means of three experiments and the corresponding standard deviations.

**Table 2 ijms-25-05048-t002:** CI values of the simultaneous treatment schedule of SAHA or AZA and cisplatin in H460 cells ^1^.

Cisplatin	0.1 µM	0.3 µM	1 µM	3 µM	10 µM
SAHA					
0.3 µM	-	1.32 ± 0.6	1.15 ± 0.4	0.67 ± 0.1	1.44 ± 0.1
1 µM	-	1.52 ± 0.5	0.68 ± 0.1	0.56 ± 0.1	1.56 ± 0.3
3 µM	-	0.71 ± 0.2	0.58 ± 0.1	0.82 ± 0.1	1.46 ± 0.6
AZA					
0.3 µM	0.98 ± 083	1.02 ± 0.43	0.49 ± 0.10	0.44 ± 0.03	0.51 ± 0.13
1 µM	0.79 ± 0.50	0.58 ± 0.38	0.38 ± 0.16	0.39 ± 0.06	0.91 ± 0.09

^1^ The H460 cells were treated simultaneously with the two drugs for 72 h. The efficacy of the treatments was evaluated through growth inhibition assays, and the data were used for the estimation of the CI. CI values between 0 and 0.7 indicate a synergistic interaction between the two drugs; CI values between 0.8 and 0.9 indicate additivity; CI values greater than 1.1 indicate antagonism. The values reported in the table represent the means ± standard deviations of three independent experiments.

**Table 3 ijms-25-05048-t003:** CI values of the pre-incubation treatment schedule of SAHA or AZA before cisplatin in H460 cells ^1^.

Cisplatin	0.1 µM	0.3 µM	1 µM	3 µM	10 µM
SAHA					
0.3 µM	-	0.98 ± 0.1	1.44 ± 0.6	0.81 ± 0.3	1.06 ± 0.1
1 µM	-	1.41 ± 0.7	1.27 ± 0.8	0.56 ± 0.1	0.83 ± 0.1
3 µM	-	0.65 ± 0.3	0.63 ± 0.3	0.63 ± 0.3	0.82 ± 0.4
AZA					
0.3 µM	0.83 ± 0.13	0.83 ± 0.07	0.30 ± 0.01	0.26 ± 0.02	0.22 ± 0.07
1 µM	0.71 ± 0.03	0.49 ± 0.09	0.36 ± 0.05	0.25 ± 0.07	0.27 ± 0.05

^1^ The H460 cells were pre-incubated for 24 h with SAHA or AZA and then co-incubated with cisplatin for 48 h. The efficacy of the treatments was evaluated through growth inhibition assays, and the data were used for the definition of the CI. CI values between 0 and 0.7 indicate a synergistic interaction between the two drugs; CI values between 0.8 and 0.9 indicate additivity; CI values greater than 1.1 indicate antagonism. The values reported in the table represent the means ± standard deviations of three independent experiments.

**Table 4 ijms-25-05048-t004:** CI values of the simultaneous treatment schedule of SAHA or AZA and cisplatin in H460/Pt cells ^1^.

Cisplatin	0.1 µM	0.3 µM	1 µM	3 µM	10 µM
SAHA					
0.3 µM	-	1.42 ± 0.5	2.34 ± 0.9	1.42 ± 0.3	1.33 ± 0.2
1 µM	-	1.39 ± 0.4	1.31 ± 0.1	1.37 ± 0.7	1.42 ± 0.2
3 µM	-	1.45 ± 0.2	1.24 ± 0.3	1.07 ± 0.2	1.39 ± 0.2
AZA					
0.3 µM	1.11 ± 0.5	1.48 ± 1.1	0.97 ± 0.2	0.46 ± 0.1	0.42 ± 0.2
1 µM	1.06 ± 0.1	0.96 ± 0.3	0.64 ± 0.1	0.23 ± 0.1	0.58 ± 0.2

^1^ The H460/Pt cells were treated simultaneously with the two drugs for 72 h. The efficacy of the treatments was evaluated through growth inhibition assays, and the data were used for the estimation of the CI. CI values between 0 and 0.7 indicate a synergistic interaction between the two drugs; CI values between 0.8 and 0.9 indicate additivity; CI values greater than 1.1 indicate antagonism. The values reported in the table represent the means ± standard deviations of three independent experiments.

**Table 5 ijms-25-05048-t005:** CI values of the pre-incubation treatment schedule of SAHA or AZA before cisplatin in H460/Pt cells ^1^.

Cisplatin	0.1 µM	0.3 µM	1 µM	3 µM	10 µM
SAHA					
0.3 µM	-	4.56 ± 4.2	2.18 ± 0.6	1.01 ± 0.1	1.15 ± 0.5
1 µM	-	2.67 ± 2.1	2.03 ± 0.9	1.02 ± 0.5	0.81 ± 0.2
3 µM	-	1.79 ± 0.7	1.16 ± 0.2	0.92 ± 0.1	0.94 ± 0.1
AZA					
0.3 µM	1.06 ± 0.3	1.62 ± 1.7	0.64 ± 0.3	0.30 ± 0.1	0.27 ± 0.1
1 µM	0.65 ± 0.1	0.59 ± 0.1	0.32 ± 0.1	0.28 ± 0.1	0.16 ± 0.1

^1^ The H460/Pt cells were pre-incubated for 24 h with SAHA or AZA and then co-incubated with cisplatin for 48 h. The efficacy of the treatments was evaluated through growth inhibition assays, and the data were used for the definition of the CI. CI values between 0 and 0.7 indicate a synergistic interaction between the two drugs; CI values between 0.8 and 0.9 indicate additivity; CI values greater than 1.1 indicate antagonism. The values reported in the table represent the means ± standard deviations of three independent experiments.

## Data Availability

The original contributions presented in the study are included in the article, further inquiries can be directed to the corresponding author.

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
