# Peer review of "KiSS-1 Modulation by Epigenetic Agents Improves the Cisplatin Sensitivity of Lung Cancer Cells"

_ijms, 2024, doi:10.3390/ijms25095048_

Round 1

Reviewer 1 Report

Comments and Suggestions for Authors

Authors have written a research article entitled “KiSS-1 modulation by epigenetic agents improves cisplatin sensitivity of lung cancer cells”. This article evaluates drug combinations between cisplatin and SAHA as well as AZA using cisplatin-sensitive H460 and -resistant H460/Pt NSCLC cells. Overall, the paper is well written however in some sections, there are some concerns which must be resolved before consideration of this manuscript. Some of the concerns are:

·       Table  2 must be formatted properly; numbers are mixed with table fonts.

·       A separate statistical analysis section must be incorporated in the manuscript.

·       Authors are suggested to include and cite the recent study related to NSCLC in introduction section: https://doi.org/10.3892/or.2018.6795  

·       In section 4.2 what type of growth-inhibition assays was performed. The methodology explained is very confusing, rectify it.

·       In figures legend, authors have not mentioned the significance level of p-value, please add this in statistical analysis section as well as in figure’s legend.

·       Authors are suggested to incorporate the recent experimental reports related to histone acetylation in line number 48-60 and cite: https://doi.org/10.1016/j.phrs.2022.106270, https://doi.org/10.1186/s12967-022-03415-6  

·       Authors must explain the criteria of selecting protein targets p53, bax and bcl2. Also justify the correlation between these proteins and KiSS-1 modulation. What type of epigenetic agents described in this study.

·       For table 1-5, authors must include the graph of respective results.

·       Discussion section must be improved by comparing the results with previous findings.

·       Authors are suggested to compare the results of p53 in discussion section (line number: 378-389) with previous reports and cite also https://doi.org/10.3892/or.2019.6964 .

·       Authors have selected only in vitro studies but in vivo studies were not reported. Explain the reason and also mention this in conclusion section for future perspective.

·       Include at least one figure exploring the mechanistic approach behind the whole concept of KiSS 1 in NSCLC.

·       Improve the English and Grammatical issues throughout the manuscript.

Comments on the Quality of English Language

Minor English and Grammatical changes are required.

Author Response

Authors have written a research article entitled “KiSS-1 modulation by epigenetic agents improves cisplatin sensitivity of lung cancer cells”. This article evaluates drug combinations between cisplatin and SAHA as well as AZA using cisplatin-sensitive H460 and -resistant H460/Pt NSCLC cells. Overall, the paper is well written however in some sections, there are some concerns which must be resolved before consideration of this manuscript.

Authors’ reply. We thank the Referee for the positive comments.

Some of the concerns are:

Table  2 must be formatted properly; numbers are mixed with table fonts.

Authors’ reply. As requested tables have been formatted.

A separate statistical analysis section must be incorporated in the manuscript.

Authors’ reply. As requested a section dedicated to the statistical analysis has been introduced in Materials and Methods (Paragraph 4.7).

Authors are suggested to include and cite the recent study related to NSCLC in introduction section: https://doi.org/10.3892/or.2018.6795

Authors’ reply. As suggested, the indicated article has been introduced. Reference #9

In section 4.2 what type of growth-inhibition assays was performed. The methodology explained is very confusing, rectify it.

Authors’ reply. We apologize for the misunderstanding and the methodology regarding growth inhibition assay, defined by counting the number of cells with cell counter, has been implemented (Paragraph 4.2)

In figures legend, authors have not mentioned the significance level of p-value, please add this in statistical analysis section as well as in figure’s legend.

Authors’ reply. The significance level of p-value has been introduced in the legends.

Authors are suggested to incorporate the recent experimental reports related to histone acetylation in line number 48-60 and cite: https://doi.org/10.1016/j.phrs.2022.106270

https://doi.org/10.1186/s12967-022-03415-6

Authors’ reply. As suggested, the indicated articles have been introduced. References #18 and 8

Authors must explain the criteria of selecting protein targets p53, bax and bcl2. Also justify the correlation between these proteins and KiSS-1 modulation. What type of epigenetic agents described in this study.

Authors’ reply. The proteins have been selected on their major role in the outcome of apoptosis. Since KiSS1 has been reported to favour apoptosis in specific cellular contexts, the modulation of KiSS1 in relation to apoptosis activation has been explored. The epigenetic agents used in this study belong to two classes: HDAC inhibitors and DNMT inhibitors. This has been clarified in paragraph 2.3 and Discussion.

For table 1-5, authors must include the graph of respective results.

Authors’ reply. As suggested, graphs of the results reported in tables have been introduced in supplementary material (Figure S1, S2, S3 and S4).

Discussion section must be improved by comparing the results with previous findings (see last paragraph of the discussion).

Authors’ reply. As requested, we have implemented the Discussion. Of note, preliminary in vivo studies have been introduced and commented (Discussion section)

Authors are suggested to compare the results of p53 in discussion section (line number: 378-389) with previous reports and cite also https://doi.org/10.3892/or.2019.6964

Authors’ reply. As suggested, the indicated reference has been introduced and commented Reference # 41.

Authors have selected only in vitro studies but in vivo studies were not reported. Explain the reason and also mention this in conclusion section for future perspective.

Authors’ reply. Manuscript has been extensively implemented and in vivo studies in mice bearing H460 and H460/Pt tumor xenografts treated with SAHA, cisplatin and the combination have been introduced. These studies are commented in the Discussion section.

Include at least one figure exploring the mechanistic approach behind the whole concept of KiSS 1 in NSCLC.

Authors’ reply. As requested by the referee, a figure summarizing the whole concept of KiSS-1 in NSCLC has been introduced (Figure 7).

Improve the English and Grammatical issues throughout the manuscript.

Authors’ reply. Typos have been corrected.

Reviewer 2 Report

Comments and Suggestions for Authors

1. Given the prevalence and mortality associated with NSCLC, and the limitations of current therapies, can epigenetic drugs and KiSS-1 offer a promising new approach for overcoming treatment challenges and improving patient outcomes?

2. How do HDAC inhibitors (like SAHA) and DNMT inhibitors (like AZA) exert their influence in NSCLC? Can we explain their impact on histone acetylation, DNA methylation, and gene expression in a way that highlights their potential therapeutic benefits for NSCLC treatment?

3. Can KiSS-1's involvement in metastasis suppression, cisplatin response, and its potential as a target in NSCLC be explored based on existing research?

4.How do epigenetic agents, such as HDAC and DNMT inhibitors, impact the anti-proliferative effect of cisplatin in NSCLC cells?

Author Response

Comments and Suggestions for Authors

1.Given the prevalence and mortality associated with NSCLC, and the limitations of current therapies, can epigenetic drugs and KiSS-1 offer a promising new approach for overcoming treatment challenges and improving patient outcomes?

Authors’ reply. We thank the Referee for this question. Yes, the pharmacological strategy studied in this manuscript is expected to improve NSCLC patient outcomes. Indeed, in spite of the advancements in the treatment of NCSLC persistent cures are difficult to achieve. The new approach investigated in the present study may provide new opportunities to explore in the clinical setting (Discussion). Future efforts are required to further investigate the possibility to enhance cisplatin response in animal models by KiSS-1 modulation. However, due to the poor bioavailability of Kisspeptins, they are not suitable for in vivo studies, but the development of KiSS-1 mimetics endowed with improved stability and bioavailability will be helpful to achieve this objective.

2.How do HDAC inhibitors (like SAHA) and DNMT inhibitors (like AZA) exert their influence in NSCLC? Can we explain their impact on histone acetylation, DNA methylation, and gene expression in a way that highlights their potential therapeutic benefits for NSCLC treatment?

Authors’ reply. We agree with the Referee. The epigenetic agents used in this study allow to target tumor cells by reactivating pathways capable of inducing cell death. The specific mechanism of action is related to the drug ability to modulate gene expression.

3.Can KiSS-1's involvement in metastasis suppression, cisplatin response, and its potential as a target in NSCLC be explored based on existing research?

Authors’ reply. Again, we agree with the Referee. Indeed, the existing research supports that KiSS-1 may act at the interface of the metastatic and apoptotic processes, thereby representing a valuable target for modulating response to clinically available antitumor agents.

  1. How do epigenetic agents, such as HDAC and DNMT inhibitors, impact the anti-proliferative effect of cisplatin in NSCLC cells?

Authors’ reply. Referee is right. The type of drug modulation, mainly a synergistic effect, is reported in Tables 2-5 and now also in Supplementary figures.

Reviewer 3 Report

Comments and Suggestions for Authors

IJMS:

Checking the blots, kindly check the journal policy about whether you need to name the lanes (wells) or not. Otherwise, the authors have provided all the raw blotting information.

Supplementary file: The authors have reported all their raw WB numerical data, thanks very much for that. 

Main text:

The role of Kiss-1 modulation by epigenetic agents may have therapeutic value

Abstract:

The authors set a background and mention how SAHA and AZA might be useful for NSCLC treatment. Also, they point to the modulation of Kiss1 by AZA. Drug combinations of cisplatin/ SAHA and cisplatin/AZA could be promising. The combinations may be triggering apoptosis. While it seems that single agents reduce apoptosis.

To be honest, the abstract could be a little clearer (the drugs used/ combinations and their effects are a little confusing and could be clearer), and implications and future directions could be mentioned too.

Intro:

As expected, the authors start by providing a background for NSCLC but this could be expanded.

The authors allude to driver alterations such as EGF-R and ALK  translocations and how novel therapeutics might be focused on targeting these alterations, while chemotherapy might still be the method of choice for NSCLC cancers without these alterations.

Apart from the low therapeutic efficacy, resistance can also ensue, and then the authors shift to epigenetic agents that may also be indicated in NSCLC treatment. The epigenetic basis of NSCLC could be expanded upon more, why is it that this genre of drugs is meant to be effective?

From there the authors expand upon HDACs and their properties. Interestingly, the authors talk about HDACs acting upon other proteins such as p53 and tubulin. 

Then intro to AZA ensues and its link to DNA damage and triggering apoptosis is mentioned.

From there the authors talk about one of the therapeutic targets of these epigenetic agents such as kiss1.  For example, Kiss1 can lead to reduced cancer invasion and metastasis. The next paragraph then talks about the role of kiss1 in NSCLC and it is thought that perhaps kiss-1 is downregulated in resistant NSCLC cells.

So the authors then outline/formulate their aims.

Overall, the intro is useful and relevant.

In Table 1, the authors outline the resistance obtained in the H460/Pt cell lines at different time points.

Table 2: CI values for cisplatin alone or in combination with different concentrations of SAHA or different concentrations of AZA, which is good. It is a suggestion, but the authors might want to show that with a heatmap (if relevant). This will give a better visual representation of the synergism indicated in line 166.

Table 3: CI values when pre-incubation with SAHA or AZA before cisplatin treatment, again a heatmap view might be a little easier to follow.

Table 4: CI values for simultaneous treatment on cisplatin-resistant cell lines. The same comment as above.

Table 5: CI values for pre-incubation with SAHA and AZA before cisplatin treatment now on cisplatin-resistant cell lines. The same comment as above.

For tables 1-5 the authors could provide some more detail as to what their data show/mean.

From here on the authors aim to understand the targets of the drugs and because it was shown that HDACi could act on p53 and apoptosis proteins, they test these markers and show that indeed the combination therapies described do enhance p53 and  BAX in both sensitive and resistant cell lines. I would suggest perhaps plotting their WB numerical data in a heatmap too (it will improve following so much data).

The data for BCL2 is a little muddled, so we can’t read into it too much.

Then in Figure 2 come other apoptosis readouts (TUNEL) and it is the higher doses of combination treatment of SAHA and Cis that are giving significant increases (in both resistant and sensitive cell lines). This suggests that perhaps the presence or absence of resistance is not wholly relevant here but rather the SAHA presence is making a difference.

Have the authors tried the same experiment for AZA and Cis? I appreciate the points you have raised in 283-285, but is there scope to test this for AZA too especially since apoptosis makers were changed in Figure 1, so why not test TUNEL too? This is important since you want to show that AZA too increases apoptosis in both resistant and sensitive cell lines and also figure 3 emphasises AZA.

Figure 3: it is the AZA combinations that show significant changes in kiss1 irrespective of cisplatin resistance status and not so much for SAHA, which is interesting. Presumably, kiss1 is a target of Aza then (but is not dependent on cisplatin resistance status? Only slightly more pronounced in resistant cell lines, correct? 

Figure 4, the authors then mimic kiss1 with a peptide and higher concentrations of it with Cis increased apoptosis in both resistant and sensitive cell lines.

Again for figures 1-4 kindly expand upon what these results mean (in the main text).

What happens to apoptosis levels once kiss1 is downregulated in cisplatin-resistant and sensitive cell lines? This could be a figure 5.

The authors could draw a model for their experiments in Figure 6.

Is kiss1 downstream to SAHA as well (I know the ELISA experiments don’t show this).

Discussion:

The authors draw evidence from the literature to shed light on their data.

In the concluding remarks, is kiss1 downtream to both HDACi and DMTi? 

The methods are useful and comprehensive.

Kudos for the interesting work.

Comments on the Quality of English Language

Some editing is required

Author Response

Comments and Suggestions for Authors

IJMS: Checking the blots, kindly check the journal policy about whether you need to name the lanes (wells) or not. Otherwise, the authors have provided all the raw blotting information.

Supplementary file: The authors have reported all their raw WB numerical data, thanks very much for that.  

Main text:

The role of Kiss-1 modulation by epigenetic agents may have therapeutic value.

Authors’ reply. We thank the Referee for the positive comments. The original Western blots include the indication of the lanes.

Abstract:

The authors set a background and mention how SAHA and AZA might be useful for NSCLC treatment. Also, they point to the modulation of Kiss1 by AZA. Drug combinations of cisplatin/ SAHA and cisplatin/AZA could be promising. The combinations may be triggering apoptosis. While it seems that single agents reduce apoptosis.

To be honest, the abstract could be a little clearer (the drugs used/ combinations and their effects are a little confusing and could be clearer), and implications and future directions could be mentioned too.

Authors’ reply. Based on Figure 2, there is a mild induction of apoptosis by SAHA alone. The abstract has been improved for clarity. Preliminary in vivo studies and future directions have been introduced in discussion.

Intro:

As expected, the authors start by providing a background for NSCLC but this could be expanded.

The authors allude to driver alterations such as EGF-R and ALK  translocations and how novel therapeutics might be focused on targeting these alterations, while chemotherapy might still be the method of choice for NSCLC cancers without these alterations.

Apart from the low therapeutic efficacy, resistance can also ensue, and then the authors shift to epigenetic agents that may also be indicated in NSCLC treatment. The epigenetic basis of NSCLC could be expanded upon more, why is it that this genre of drugs is meant to be effective?

From there the authors expand upon HDACs and their properties. Interestingly, the authors talk about HDACs acting upon other proteins such as p53 and tubulin. 

Then intro to AZA ensues and its link to DNA damage and triggering apoptosis is mentioned.

From there the authors talk about one of the therapeutic targets of these epigenetic agents such as kiss1.  For example, Kiss1 can lead to reduced cancer invasion and metastasis. The next paragraph then talks about the role of kiss1 in NSCLC and it is thought that perhaps kiss-1 is downregulated in resistant NSCLC cells.

So the authors then outline/formulate their aims.

Overall, the intro is useful and relevant.

Authors’ reply. We thank the Referee for the positive comment regarding the introduction. We have implemented the discussion in relation to the potential interest of epigenetic agents as antitumor therapeutics. In this regard, the manuscript has been importantly implemented. Preliminary in vivo studies carried out in mice bearing H460 and H460/Pt tumor xenografts have been introduced. These studies are commented in the Discussion and Conclusion sections. Moreover, future directions have been discussed.

In Table 1, the authors outline the resistance obtained in the H460/Pt cell lines at different time points.

Table 2: CI values for cisplatin alone or in combination with different concentrations of SAHA or different concentrations of AZA, which is good. It is a suggestion, but the authors might want to show that with a heatmap (if relevant). This will give a better visual representation of the synergism indicated in line 166.

Table 3: CI values when pre-incubation with SAHA or AZA before cisplatin treatment, again a heatmap view might be a little easier to follow.

Table 4: CI values for simultaneous treatment on cisplatin-resistant cell lines. The same comment as above.

Table 5: CI values for pre-incubation with SAHA and AZA before cisplatin treatment now on cisplatin-resistant cell lines. The same comment as above.

For tables 1-5 the authors could provide some more detail as to what their data show/mean.

Authors’ reply. We thank the Referee for the suggestion. The heat maps have been provided in the Supplementary material (Table S1). According to the Chou-Talalay method, the effect of drug combinations was evaluated by exposing the cells to different concentrations of drugs. By applying the Calcusyn software (Biosoft, Cambridge, UK), this method assigns a combination index (CI) value to the drug combination. CI lower than 0.85–0.90 indicate synergistic drug interactions, conversely CI values higher than 1.20–1.45 or around 1 reflect antagonism or additive activity, respectively. The methodology is described in Materials and Methods section.

From here on the authors aim to understand the targets of the drugs and because it was shown that HDACi could act on p53 and apoptosis proteins, they test these markers and show that indeed the combination therapies described do enhance p53 and  BAX in both sensitive and resistant cell lines. I would suggest perhaps plotting their WB numerical data in a heatmap too (it will improve following so much data).

The data for BCL2 is a little muddled, so we can’t read into it too much.

Authors’ reply. We agree with the Referee regarding the modulation of BCL2 levels. According to our experience, BCL2 levels are never modulated in a strong manner like BAX and p53. Anyway, we have chosen to show the results we have obtained. The heat map is reported in Supplementary material (Table S2).

Then in Figure 2 come other apoptosis readouts (TUNEL) and it is the higher doses of combination treatment of SAHA and Cis that are giving significant increases (in both resistant and sensitive cell lines). This suggests that perhaps the presence or absence of resistance is not wholly relevant here but rather the SAHA presence is making a difference.

Authors’ reply. We agree with the Referee. The drug combination is synergistic in both sensitive and resistant cells. However, the synergism is more evident for the cisplatin-resistant variant.

Have the authors tried the same experiment for AZA and Cis? I appreciate the points you have raised in 283-285, but is there scope to test this for AZA too especially since apoptosis makers were changed in Figure 1, so why not test TUNEL too? This is important since you want to show that AZA too increases apoptosis in both resistant and sensitive cell lines and also figure 3 emphasises AZA.

Authors’ reply. We agree with the Referee. The apoptotic induction of the combination cispatin with AZA has been introduced. Text has been modified accordingly (Paragraph 2.3 and Figure 3)

Figure 3: it is the AZA combinations that show significant changes in kiss1 irrespective of cisplatin resistance status and not so much for SAHA, which is interesting. Presumably, kiss1 is a target of Aza then (but is not dependent on cisplatin resistance status? Only slightly more pronounced in resistant cell lines, correct? 

Authors’ reply. Yes, the Referee is right. The original Figure 3 is now the Figure 4.

Figure 4, the authors then mimic kiss1 with a peptide and higher concentrations of it with Cis increased apoptosis in both resistant and sensitive cell lines.

Again for figures 1-4 kindly expand upon what these results mean (in the main text).

Authors’ reply. A comment to each figure has been added. The original Figure 4 is now the Figure 5.

What happens to apoptosis levels once kiss1 is downregulated in cisplatin-resistant and sensitive cell lines? This could be a figure 5.

Authors’ reply. We thank the Referee for the suggestion. When running loss of function studies in H460 cells by transfection of a pool of siRNAs, we did not observe a change of apoptosis levels (see figure below). We think that this phenomenon may be dependent either on the activation of compensatory pathways upon KiSS-1 knock-down or on the fact that the response of the drugs involves too many cellular events not necessarily depending on KiSS-1 silencing. We have chosen to cite these results as not shown (Discussion).    

The authors could draw a model for their experiments in Figure 6.

Is kiss1 downstream to SAHA as well (I know the ELISA experiments don’t show this).

Discussion:

The authors draw evidence from the literature to shed light on their data.

In the concluding remarks, is kiss1 downtream to both HDACi and DMTi? 

The methods are useful and comprehensive

Authors’ reply. Yes, KiSS-1 is downstream of both HDACi and DNMTi because it is upmodulated following exposure to SAHA and AZA (see ref. 21 and 39). As requested also by the Referee #1, an additional Figure (Figure 7) has been introduced to describe a possible cellular model.  

Round 2

Reviewer 1 Report

Comments and Suggestions for Authors

Authors have revised the manuscript as per the suggestions, thus it can be considered for publication in its present form.

Comments on the Quality of English Language

Proof read is required

Reviewer 3 Report

Comments and Suggestions for Authors

The authors have attempted to address my comments and have made meaningful changes. The addition of the heatmaps to supplementary has been useful and will make tracking synergism and chnages to proteins levels in western blotting much easier. Also, the mechanistic figure is useful since it really sums up the evidence.

Only a minor point that can be addressed after acceptance is the difference between decimal points and commas when writing numbers. This is the new supplementary data in which for example a number like 1,32 is reported this way while I think the authors mean 1.32 in that .32 are decimal numbers. I do see this a lot with European colleagues and for them comma seperates the number from the decimals. Please refer to the journal guidelines.

Comments on the Quality of English Language

Minor changes required